# *Helping or Herding?* 🐑
# Reward Model Ensembles Mitigate but do not Eliminate Reward Hacking

**Jacob Eisenstein**[1,*]          **Chirag Nagpal**[2,*]          **Alekh Agarwal**[2,*]

**Ahmad Beirami**[1]      **Alex D'Amour**[1]      **DJ Dvijotham**[1]      **Adam Fisch**[1]

**Katherine Heller**[2]      **Stephen Pfohl**[2]      **Deepak Ramachandran**[1]      **Peter Shaw**[1]

**Jonathan Berant**[1,*]

1. Google DeepMind
2. Google Research
* Core contributors
`reward-ensembles-helping-or-herding@google.com`

## Abstract

Reward models play a key role in aligning language model applications towards human preferences. However, this setup creates an incentive for the language model to exploit errors in the reward model to achieve high estimated reward, a phenomenon often termed *reward hacking*. A natural mitigation is to train an ensemble of reward models, aggregating over model outputs to obtain a more robust reward estimate. We explore the application of reward ensembles to alignment at both training time (through reinforcement learning) and inference time (through reranking). First, we show that reward models are *underspecified*: reward models that perform similarly in-distribution can yield very different rewards when used in alignment, due to distribution shift. Second, underspecification results in overoptimization, where alignment to one reward model does not improve reward as measured by another reward model trained on the same data. Third, overoptimization is mitigated by the use of reward ensembles, and ensembles that vary by their *pretraining* seeds lead to better generalization than ensembles that differ only by their *fine-tuning* seeds, with both outperforming individual reward models.[1] However, even pretrain reward ensembles do not eliminate reward hacking: we show several qualitative reward hacking phenomena that are not mitigated by ensembling because all reward models in the ensemble exhibit similar error patterns.

## 1   Introduction

To align machine learning systems with human preferences, it is common to use *reward models (RMs)*, which are finetuned on preference annotations to score potential outputs by how likely they are to be preferred by human raters (Christiano et al., 2017; Stiennon et al., 2020). There are many ways to use RMs to align policy models: they can act as training signals in reinforcement learning (Christiano et al., 2017; Stiennon et al., 2020), they can select examples for further imitation learning (Gulcehre et al., 2023; Liu et al., 2023; Dong

---

[1]These pretrains are available at `https://github.com/google-deepmind/reward-ensembles`.

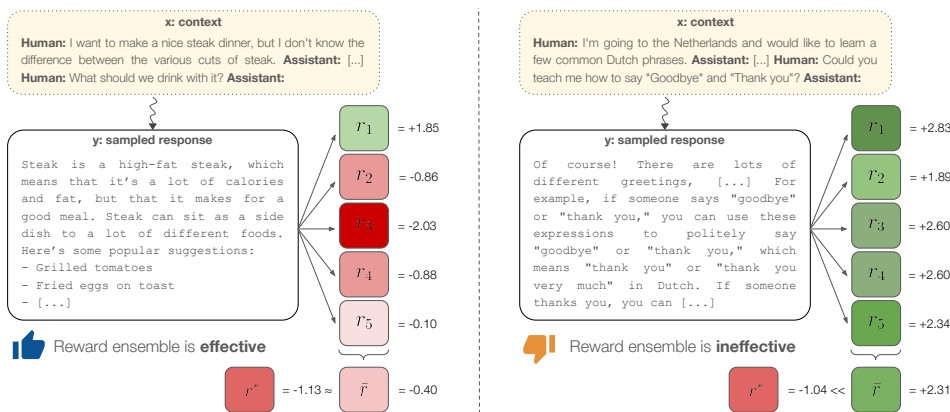

Figure 1: Left: RM ensembles can attenuate errors made by individual RMs, in this case the positive $r_1$ for this off-topic response from the policy model $\pi(y \mid x)$, which gets a low true reward ($r^*$). Right: insufficiently diverse RMs unanimously rate this overly-verbose and non-responsive reply from $\pi(y \mid x)$ as positive, but it too gets a low true reward. Both examples are real outputs and rewards (represented as normalized Z-scores) from RMs trained on the HELPFULNESS data of Bai et al. (2022).

et al., 2023; Touvron et al., 2023), or they can be applied at inference time to steer the output distribution toward higher expected reward (e.g., Yang & Klein, 2021; Gao et al., 2023). Such procedures create a semi-adversarial dynamic, in which the language model can achieve high reward by exploiting errors in the RM. Furthermore, while the RM is trained on a fixed set of human preference data, the process of alignment shifts the distribution of its inputs, increasing the likelihood of such errors. This phenomenon where the policy language model exploits reward model errors is often termed *reward hacking* (Amodei et al., 2016), *reward gaming* (Skalse et al., 2022; Pang et al., 2023), or *reward over-optimization* (Gao et al., 2023).

Reward hacking has been investigated from several perspectives in prior work (e.g., Krakovna et al., 2020; Skalse et al., 2022; Pan et al., 2022). Bai et al. (2022) used reinforcement learning with human feedback (RLHF) and trained two RMs on non-overlapping splits of preference data, using one to drive alignment, and the other to measure the quality of the outputs. They find that RLHF increases performance according to both the driver and measurement models, but that a performance gap emerges as the policy is allowed to diverge from the initial distribution. However, both RMs were built on base models trained on the same *pretraining* data, which, as we will show, limits their diversity (as hypothesized by Gleave & Irving (2022)) and thus may understate the effect of reward hacking. Other work has simulated the relationship between a "true" reward and a learned proxy, showing that it is possible to over-optimize the proxy to such an extent that the true reward starts to decrease (Gao et al., 2023; Coste et al., 2023), e.g. by exploiting spurious correlations in reward model training data (Pang et al., 2023).

In this work, we first analyze RM distribution shift from the perspective of *underspecification* (D'Amour et al., 2022), which occurs when a machine learning pipeline yields reliable performance on held-out data from the training distribution, but variable performance on out-of-distribution data. When applied to learning RMs from human preference data, we show that RMs that agree in-distribution often disagree after alignment-induced policy distribution shifts. Furthermore, such disagreements are more pronounced when the RMs are built on different *pretrainings*, even when that difference is induced merely by varying the pretraining random seed. These disagreements become increasingly severe when evaluated on outputs of a policy model that has been aligned to a specific RM. This occurs both when using RMs in RLHF, as well as when using an inference-time alignment procedure, best-of-$n$ reranking, where $n$ samples are drawn from the policy and then reranked with a RM.

Motivated by these findings, we systematically investigate reward model ensembles as a possible remedy for reward hacking. Assuming different models err in different ways,

ensembling can leverage reward uncertainty across the ensemble during alignment (see Figure 1, Left). We explore several techniques for aggregating scores across the ensemble, e.g., taking the median score as a robust estimate of the true reward of the policy. We also consider two types of ensembles: *pretrain ensembles*, where different members of the ensemble differ in the random seed used during the pretraining phase, and *finetune ensembles*, where members differ only in the random seed used during finetuning. These ensembles are then evaluated across several types of policies and preference annotations: dialogue preferences for a helpful assistant (Bai et al., 2022), summarization quality (Stiennon et al., 2020), and whether a summary is grounded in its source text (Roit et al., 2023).

We find that ensembles are significantly more robust than individual reward models, and that pretrain ensembles are particulary effective. However, reward ensembles are still susceptible to reward hacking when all members of the ensemble share similar error patterns, which in turn propagate to the ensemble (see Figure 1, Right). This is exploited and amplified by policy optimization: for example, summarization models produce outputs that are too short when tuned for factuality, but too verbose when tuned for summarization quality; assistant models overuse formulaic answer formats when tuned for helpfulness.

**Recent and concurrent related work**  Coste et al. (2023) argue that reward model ensembles effectively mitigate reward hacking. Our work shares a similar research question, but differs in several ways, leading to more nuanced conclusions: we investigate both pretrain and finetune ensembles; we use human-annotated preference data rather than synthetically-generated labels; and we analyze several real reward hacks, showing that they are not prevented by ensembles. Subsequent work has explored efficient RM ensembles through low-rank adaptation (e.g., Zhai et al., 2023) and weight averaging (Ramé et al., 2024), but exclusively for finetune ensembles. More generally, we can try to optimize the policy against the worst-case reward in some analytically-computed uncertainty set (Zhu et al., 2023; Zhang et al., 2024), but this too accounts only for uncertainty due to the finite sample of preference pairs and not in the representations made available through pretraining.

## 2 Preliminaries

We now briefly review how reward models are trained (§2.1) and how they are used for alignment (§2.2). We then describe our setup for studying reward hacking (§2.3).

### 2.1 Reward Model Training

RMs are typically trained from *preference data*, $(x, y^+, y^-) \in D$, where $y^+$ is preferred over $y^-$ for prompt $x$. Under the Bradley-Terry model (Bradley & Terry, 1952), the probability that response $y_2$ is preferred over $y_1$ given a reward function $r$ and a prompt $x$ is $p(y_1 \prec y_2 \mid x) = \sigma(r(x, y_2) - r(x, y_1))$, where $\sigma(\cdot)$ is the sigmoid function. Given a dataset of preferences, the maximum-likelihood objective is,

$$\mathcal{J}(r) = \mathbb{E}_{(x,y^+,y^-)\sim D} \left[ \log p(y^- \prec y^+ \mid x) \right]. \tag{1}$$

The Bradley-Terry model is underdetermined: for any RM $r^*$, there is a set of equivalent models, $r'(x, y) = r^*(x, y) + C(x)$ where $C(x)$ is a prompt-dependent constant, such that $\mathcal{J}(r^*) = \mathcal{J}(r')$. This is problematic for ensembling: if different RMs choose different values for $C(x)$, then order statistics like median and minimum are meaningless. We therefore modify the objective by regularizing the sum of rewards per preference pair towards zero:

$$\mathcal{J}_{\text{reg}}(r) = \mathcal{J}(r) + \eta \cdot \mathbb{E}_{(x,y^+,y^-)\sim D} \left[ (r(x, y^+) + r(x, y^-))^2 \right], \tag{2}$$

where $\eta$ is a small positive value, thereby resolving the issue of underdetermination.

Note that RMs can also be trained from "pointwise" data, such as toxicity or factuality annotations on individual examples (Yang & Klein, 2021; Roit et al., 2023). Such RMs are not underdetermined and so can be aggregated without adjustment.

## 2.2 Aligning Language Models using Reward Models

**Best-of-$n$ reranking (BoN)** is an inference-time alignment strategy, where given a prompt $x$, we sample $n$ generations $y_1, \ldots, y_n$ from a *policy* language model $\pi(y \mid x)$ and return the generation with highest reward: $y* = \arg\max_{y_k \in \{y_1, \ldots, y_n\}} r(x, y_k)$. The Kullback–Leibler (KL) divergence of BoN from the initial policy is upper bounded by $\log n - \frac{n-1}{n}$ (Beirami et al., 2024). BoN tends to outperform more elaborate alignment techniques like RLHF in the low-KL regime (Gao et al., 2023), albeit with the cost of generating multiple samples at inference time.

**Reinforcement Learning from Human Feedback (RLHF)** is an online reinforcement learning method that trains a policy language model $\pi$ to maximize expected reward, while staying close to an initial policy, $\pi_{\text{sft}}$, which is typically finetuned on supervised data (prompt-output pairs). Distance from the initial policy is measured with KL divergence, which leads to the regularized objective

$$\max_{\pi} \; \mathbb{E}_{\substack{x \sim \rho \\ y \sim \pi}}[r(x, y)] - \lambda \text{KL}(\pi \| \pi_{\text{sft}}), \tag{3}$$

where $r$ is an RM, $\rho$ is a distribution over prompts, and $\lambda$ is a hyperparameter. Typically, this objective is optimized using PPO (Schulman et al., 2017), which we also use in this work.

The KL penalty in eq. (3) can limit reward hacking by keeping the policy close to $\pi_{\text{sft}}$. However, KL regularization does not directly address reward model errors, and in particular does not address RM distribution shift when the preference annotations are not sampled from $\pi_{\text{sft}}$ (as is generally the case). Furthermore, we *want* to diverge from the reference policy when we can be confident of improving the expected reward. As we will show empirically, KL regularization is complementary to the development of more robust reward models, which yields pareto improvements in the reward-KL tradeoff.

## 2.3 Experimental Setup

**Datasets**   We consider three tasks. Example instances are provided in Table 12.

- TL;DR: A summarization benchmark where authors summarize their own reddit posts (Völske et al., 2017; Stiennon et al., 2020), frequently used in research on RLHF and related methods (Rafailov et al., 2023; Zhao et al., 2023).
- HELPFULNESS: A popular dialogue benchmark (Bai et al., 2022), where given a partial conversation between a human and a digital assistant the goal is to complete the next assistant turn (Bai et al., 2022; Rafailov et al., 2023).[2]
- XSUM/NLI: A summarization benchamark where a policy model trained on XSum (Narayan et al., 2018) is finetuned to generate summaries that are factually consistent with the source document (Roit et al., 2023).

**Training reward models**   To examine the effect of pretraining, we pretrain five T5 models from scratch at the base (220M parameters), large (770M), and XL (3B) scales, using the standard denoising objective over the C4 corpus (Raffel et al., 2020). The pretrained checkpoints differ only in their random seed, which controls parameter initialization and the sample from the pretraining data.

For each task, we finetune each pretrained model five times using different random seeds. In TL;DR and HELPFULNESS we use the aforementioned preference data. For XSUM/NLI, we finetune *pointwise* natural language inference (NLI) models on the ANLI dataset (Nie et al., 2020). This yields 25 RMs per task and scale (5 pretrain × 5 finetune), making it possible to evaluate the effect of pretraining and finetuning on underspecification (§3) by constructing ensembles that differ in either pretrain or finetune seed (§4).

---

[2]We use the **base** dataset (44K examples), where responses are generated from a 52B context-distilled LM, and split the training set into two: half for training the RM, and half for the policy.

| Model Size | TL;DR | HELPFULNESS | XSum/NLI |
|---|---|---|---|
| T5-BASE | $65.8 \pm 0.3$ | $66.7 \pm 0.7$ | $86.7 \pm 0.9$ |
| T5-LARGE | $69.3 \pm 0.7$ | $68.5 \pm 0.4$ | $88.3 \pm 1.2$ |
| T5-XL | $71.4 \pm 0.8$ | $69.2 \pm 0.6$ | $91.3 \pm 0.5$ |
| T5-XXL | $79.5$ | $71.5$ | $92.9$ |

Table 1: Mean in-distribution accuracy of 25 RMs on validation data for TL;DR, HELPFULNESS, and XSUM/NLI. Standard deviation, indicated with $\pm$, is small in-distribution. The single T5-XXL RM is used for evaluation purposes only.

**Alignment strategy** We use the publicly available T5-large (Raffel et al., 2020) as a policy for the summarization tasks. For HELPFULNESS, which requires substantial background knowledge, we use the instruction-tuned PALM-2-XXS model (Anil et al., 2023). Before alignment, we create a finetuned policy $\pi_{\text{sft}}$ through supervised finetuning: We finetune on annotated summaries from TL;DR and XSUM/NLI for the corresponding tasks, and on the preferred responses, $(x, y^+)$, from the preference data in HELPFULNESS.

In **BoN** reranking, we rerank sampled output sets of size $n \in \{2^1, 2^2, \ldots, 2^5\}$ for HELP-FULNESS and $\{2^1, \ldots, 2^6\}$ for TL;DR. Larger sets lead to higher reward at a cost of more expensive inference and larger deviation from $\pi_{\text{sft}}$. In **RLHF**, we obtain a trade-off between the KL from $\pi_{\text{sft}}$ and the expected reward by training multiple times, varying the value of $\lambda$. Low values of $\lambda$ correspond to high KL and high reward, while high values of $\lambda$ entail low KL and low reward. For each value of $\lambda$ we train roughly to convergence using a predetermined fixed number of steps (all hyperparameter values, including $\lambda$ and the number of steps, are in Appendix C). Coste et al. (2023) trade-off KL and reward by tracking their values during training; however, for any particular value of KL the reward might still be underoptimized during training (i.e., there can exist a different policy $\pi(y \mid x)$ with better reward, but the same $\text{KL}(\pi(y \mid x) \| \pi_{\text{sft}}(y \mid x))$, which can be found with longer training).

**Evaluation** The aligned policies are autoevaluated by two metrics: reward and win rate. Similar to past work (Gao et al., 2023; Coste et al., 2023), we use a larger RM to evaluate the generalization of models trained with a smaller RM. We train a T5-XXL RM by taking the publicly available T5-XXL (Raffel et al., 2020) and finetuning it as described above. As shown in Table 1, T5-XXL outperforms the best T5-XL model on each task. We report both average reward from the T5-XXL evaluator as well as *win rate*, which is the fraction of prompts for which the response sampled from the aligned policy $\pi$ has higher reward compared to $\pi_{\text{sft}}$.

Because the T5-XXL autoeval model is trained on the same data as the smaller T5 RMs, their errors might be correlated. For this reason, we also compute *win rate* according to a prompted PALM-2-Large model, which was not exposed to the reward training data but was instruction-tuned on Flan (Wei et al., 2022). Given a prompt $x$, we sample a response $y_{\text{sft}}$ from $\pi_{\text{sft}}$ and $y_{\text{rlhf}}$ from $\pi$. We then ask PALM-2 which response is better, using a hand-engineered prompt proposed by Rafailov et al. (2023). To avoid position bias, we run PALM-2 on the two possible orderings $(y_{\text{sft}}, y_{\text{rlhf}})$ and $(y_{\text{sft}}, y_{\text{rlhf}})$, sample $K = 8$ outputs for each order and determine the winner on this prompt through majority voting. This style of evaluation has become popular (Dubois et al., 2023; Singhal et al., 2023) and was shown to correlate well with human judgements (Rafailov et al., 2023).

## 3 Underspecification in Reward Models

We begin by demonstrating that the out-of-distribution performance of individual RMs is underspecified. Table 1 shows the average in-distribution accuracy across the 25 different RMs, together with the standard deviation. The low standard deviation highlights that all 25 reward models achieve similar performance in-distribution. But when we move to out-of-distribution data, the models diverge. Figure 2 shows the expected reward achieved by BoN as a function of the number of sampled candidates, $n$. The dotted green line

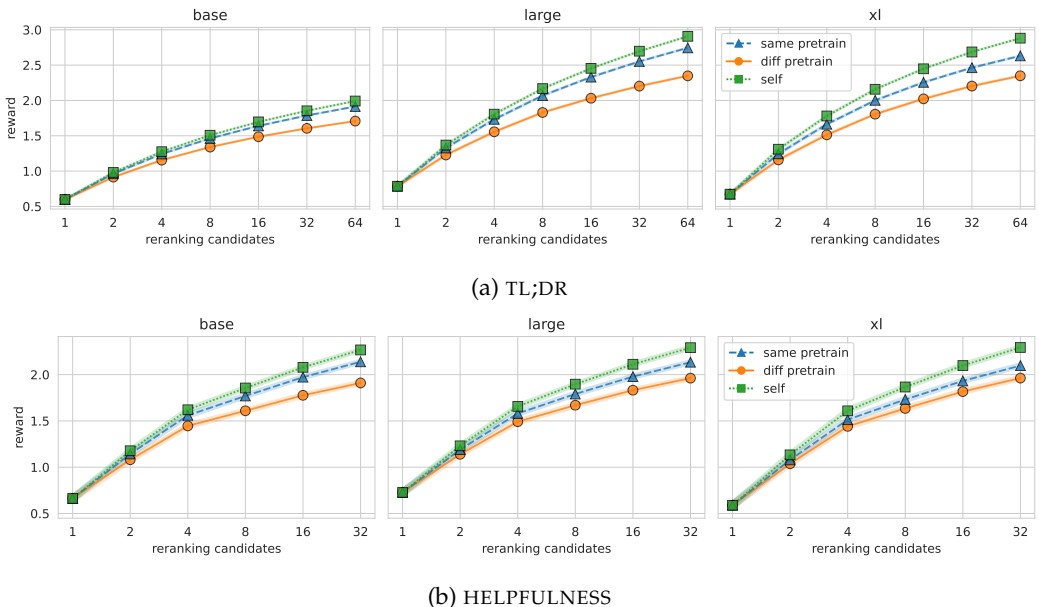

(a) TL;DR

(b) HELPFULNESS

Figure 2: Average reward of the best-of-*n* output, as judged by: the same RM used for ranking (*self*); RMs fine-tuned from the same pretrain as the ranker (*same pretrain*); RMs fine-tuned from different pretrains from the ranker (*diff pretrain*). The RMs that do not share a pretrain with the ranker regard the ranker's preferred outputs as significantly worse.

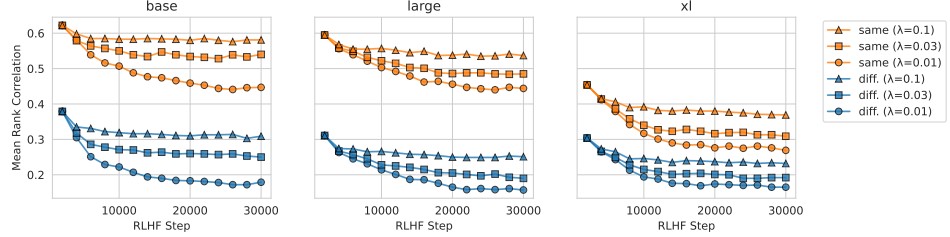

Figure 3: Rank correlation of reward scores for TL;DR RMs that share a pretraining seed and models that do not. RLHF alignment increases disagreements between RMs (lower correlation), particularly at low values of $\lambda$ and for RMs that do not share a pretrain.

shows the expected reward of the top-ranked output according to the reranker itself, while the dashed blue line shows the expected reward of the same output according to RMs that share a pretrain seed. The solid orange line shows the expected reward according to RMs that do not share a pretrain seed. Unsurprisingly, the reranker scores its own top outputs more favorably than the other RMs do. However, the reranker's outputs are scored significantly *less* favorably by RMs which do *not* share a pretrain with the ranker. RMs that share a pretrain seed with the ranker model overestimate the true reward of the top-ranked output—suggesting that finetune ensembles are not sufficiently diverse because of the shared pretraining state of each of the ensemble's members. Notably, this gap does *not* disappear with scale, as shown in Figure 2.

Moving to alignment, differences in estimated rewards induce different policies from the BoN strategy. Figure 8 (in the appendix) shows the effects on agreement of the top-ranked summary when RMs do (crosses) or do not (circles) share pretraining seeds. Different RMs tend to produce different 1-best outputs, and these differences are linked to the pretraining seed: for example, two RMs from different pretrains will choose a different best-of-16 output more than half the time for both TL;DR and HELPFULNESS and in all scales.

Figure 3 analyzes the evolution of agreement of the estimated reward scores when performing RLHF on TL;DR. For each policy model, we sample five completions for each prompt

in the validation set at 2000 step intervals during RLHF. We then measure how well pairs of RMs agree on the ranking of these completions, using Spearman rank correlation. The correlation is averaged across all pairs of reward models that do and do not share the same pre-training seed; both sets of RMs include the one used to drive RLHF. As shown in the figure, RLHF decreases the average correlation, particularly at low levels of regularization. Furthermore, agreement is substantially lower for pairs of reward models that do not share the same pretraining seed. This supports our conclusion that reward modeling under alignment-induced distribution shift is underspecified by the preference annotations.

Overall, our analysis demonstrates that (1) different RMs tend to disagree on out-of-distribution data, particularly when the RMs have different pretraining seeds; (2) this propagates to the trained policy model, in the sense that the resulting policy is highly tuned to the preferences of the specific RM used to drive it; and (3) as a result, the disagreement between RMs tends to increase during alignment. These findings suggest that reward model ensembles might mitigate reward hacking, which we turn to next.

## 4 Reward Model Ensembles

A natural mitigation to reward model underspecification is to ensemble multiple RMs, under the assumption that different models will have different errors. Aggregating over the scores of the ensemble members will help when some of the ensemble members erroneously assign high reward to a bad output.

**Building reward model ensembles**  Given a set of RMs $\mathcal{M}$, we define the reward of the ensemble to be $\bar{r}(x, y) = \text{agg}(\{r_m(x, y)\}_{m \in \mathcal{M}})$, with agg indicating an aggregation function (Dietterich, 2000; Lakshminarayanan et al., 2017; Raffel et al., 2020; Zaidi et al., 2021). Intuitively, the aggregation function should be conservative, and return a lower score when there is disagreement between the ensemble members. We consider the following simple aggregation functions: MEAN, MEDIAN, and MEAN_MINUS_STD, which subtracts the standard deviation of the reward from the mean to penalize high variance. We also experiment with MIN, but overall find it to be inferior to the alternatives.

We evaluate two types of reward ensembles: *pretrain ensembles*, where each member is pretrained using a different random seed,[3] and *finetune ensembles*, where members share the same pretraining seed, but use a different seed when finetuned on the reward data. In all cases the ensemble contains five individual RMs. Pretrain ensembles are expensive to train, but are more diverse and hence likely to lead to a more robust reward estimate. (Gleave & Irving (2022) reported negative results when using reward ensembles and hypothesized this is due to members sharing the same underlying pretrained model.)

**Evaluation**  We now evaluate RM ensembles across tasks. Figure 4 shows the results in best-of-*n* reranking, as measured by an XXL-scale fine-tuned RM. Pretrain ensembles consistently improve performance over individual RMs, especially for higher values of *n* for both TL;DR and HELPFULNESS. Finetune ensembles, conversely, improve performance in some cases and are comparable in others. For example, on TL;DR a pretrain ensemble with the MEAN aggregator achieves a win rate of 90% over the SFT outputs at the XL scale, while the win rate of a finetune ensemble with the same aggregator is 87.3%. The win rate of the average individual XL-scale RM is 85.3% (see Table 6). For visual clarity, in Figure 4 we show only two aggregators: MEAN and MEAN_MINUS_STD; see Appendix A for the other aggregators. In general, differences between aggregators are small, with MEAN usually performing at, or near, the top. More conservative aggregators (MIN and MEAN_MINUS_STD) come out slightly ahead of MEAN at the smaller scales on TL;DR, suggesting that high variance may be a bigger issue in this setting.

Figure 5 shows the KL-reward trade-off of ensemble RMs in RLHF for TL;DR and HELPFULNESS (evaluated with the finetuned T5-XXL model). In such plots, a better model

---

[3]Pretraining does not complete a single epoch over the pretraining data, and thus the data observed by each member of a pretrain ensemble is different (but sampled from the same distribution).

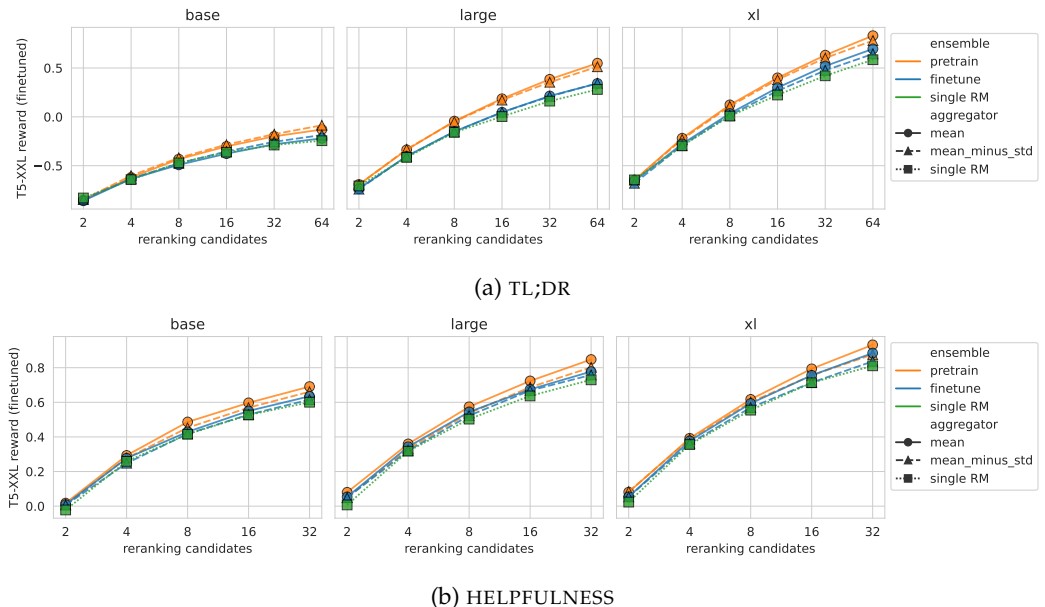

(a) TL;DR

(b) HELPFULNESS

Figure 4: In BoN reranking, pretrain ensemble RMs significantly improve output quality, as measured by a T5-XXL autoeval model. Full numerical results are in Appendix A.

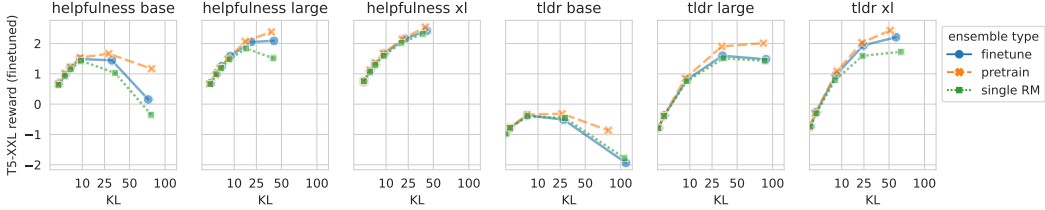

Figure 5: In RLHF, pretrain ensemble RMs lead to more favorable reward-KL tradeoffs, as judged by a T5-XXL autoeval model. Each point corresponds to training to convergence at a particular value of $\lambda$. We show MEDIAN ensembles here; for others see Appendix B.

improves reward and/or reduces KL from the SFT policy (Gao et al., 2023; Coste et al., 2023). As with BoN alignment, pretrain ensembles consistently outperform finetune ensembles as well as the average individual RM. For clarity, the figure presents results only for the MEDIAN aggregator for clarity, but these conclusions are supported by full numerical results across aggregators, shown in Appendix B. RLHF leads to much higher KL values than BoN. Consequently, we witness explicit reward hacking, in which the T5-XXL rewards decrease even as the RLHF objective improves. This happens most prominently for individual models, in many cases for finetune ensembles, and most rarely for pretrain ensembles—where T5-XXL reward scores decrease only when RLHF uses a T5-Base RM. Thus, our experiments on real data yield more negative conclusions than Coste et al. (2023) about the potential of ensembles to eliminate reward overoptimization.

As the T5-XXL autoeval model is trained on the same distribution as the RMs used for alignment, it may overstate their performance. Thus, we also use a zero-shot autoeval model, PALM-2-Large (see Section 2.3). Because this evaluation is computationally expensive, we apply it to only the largest-scale RMs (XL). As shown in Figure 6, ensemble RMs achieve higher win rates on both tasks and with both alignment techniques. For best-of-$n$, pretrain ensembles get significantly higher win rates on TL;DR at $n = 64$ ($p < .001$ by a permutation test); on HELPFULNESS the differences between ensembling techniques are not significant at $n = 32$. On both tasks, single RMs are significantly worse, $p < .001$. For RLHF, pretrain ensembles achieve better or equal win rates at lower KL divergence

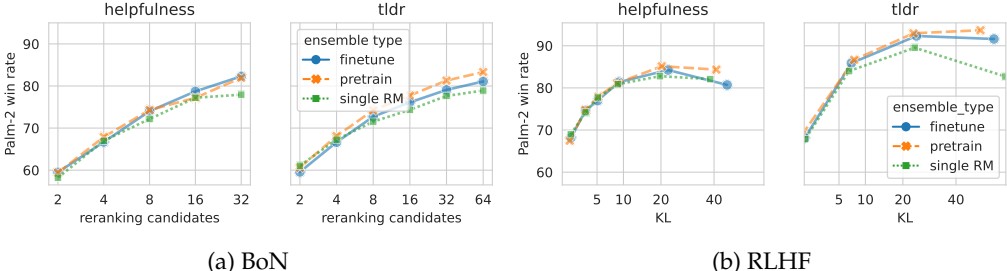

(a) BoN                                          (b) RLHF

Figure 6: Using a prompted autoevaluator (PALM-2-Flan), ensemble RMs get significantly better win rates on both TL;DR and HELPFULNESS. Here all RMs are XL-scale.

from the reference policy, with particularly strong performance on HELPFULNESS. Overall, these results mirror the T5-XXL evaluation, with one interesting difference: the PALM-2 autoeval reveals more reward hacking for RLHF, where win rate decreases with KL. This suggests that fine-tuned autoevaluators can overestimate performance when trained on the same preference data as the alignment RMs.

Figure 9 (in the appendix) shows RLHF results for XSUM/NLI. Here ensembles offer relatively small improvements, and there is little difference between pretrain and finetune ensembles. We conjecture this is because XSUM/NLI optimizes specifically for factuality. This allows all models to find simple and similar strategies that lead to high reward (namely, emitting short responses), and thus ensembling does not lead to large gains in performance.

## 5    When do Reward Model Ensembles Fail?

Ensembles improve performance according to automatic evaluation metrics, but does this mean that reward overoptimization is solved? We now investigate five specific "reward hacks", and show that they are not prevented by ensembling, because all members agree that these hacks improve the quality of the outputs.

To demonstrate this, we manually identify five qualitative distribution shifts in which the post-RLHF policies differ dramatically from both the reference policy and the preference annotations. Figure 7 and Figure 10 (in the appendix) show the results of this analysis on all benchmarks with a T5-large RM. The x-axis corresponds to the number of RLHF training steps, and the y-axis is a statistic of interest (e.g., average output length). We plot the statistic for the pretrained ensemble (using MEAN as a representative aggregation function) and for its members. For TL;DR and HELPFULNESS, where the reward model is trained on preference data, we show the statistic value on the preference data validation set, conditioned on the label 'Preferred' or 'Rejected'.

- For HELPFULNESS (Figure 7a), outputs tend to be in a list format, which we capture with a regular expression. The fraction of outputs that have this pattern increases to roughly 50% for three members of the ensemble, and for the ensemble itself. We do not detect this tendency in the preference data: the fraction of outputs that matches this format is roughly 8% for preferred and rejected responses.

- For TL;DR (Figure 7b, Figure 10b), RLHF leads to longer summaries (Singhal et al., 2023) and more extractive outputs, i.e., more copying from the input. Length in characters grows substantially for the ensemble and its members, where for the ensemble, length increases by a factor of two. On the preference data, preferred responses are slightly longer than rejected responses, but much shorter than outputs post-RLHF. We also compute the longest common subsequence (in characters) between the document and the summary and find that it doubles after RLHF with an ensemble RM. Although preference data shows a slight preference for copying, this preference is dramatically amplified by RLHF.

- For XSUM/NLI (Figure 7c, Figure 10c), training for factuality makes summaries shorter and less specific, as measured by omission of numerical values. All members of the

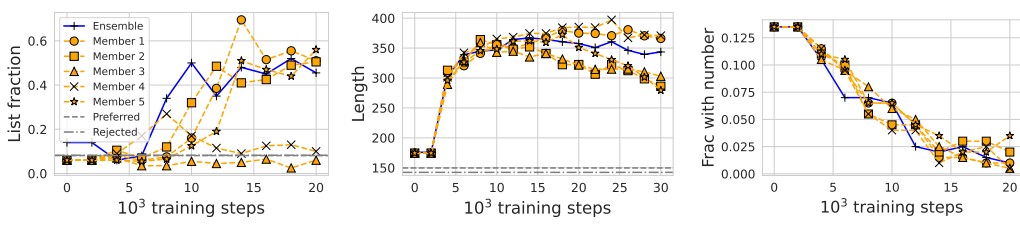

(a) HELPFULNESS: fraction of answers containing lists

(b) TL;DR: output length

(c) XSUM/NLI: fraction of numerical tokens in the output.

Figure 7: Limitations of reward model ensembles. The x-axis is RLHF steps, the y-axis plots different statistics of the average validation output at that step, and the curves correspond to the pretrain ensemble (solid blue) and its members (dashed orange). For preference data, we plot the same statistics conditioned on the preference data label (*Preferred* vs. *Rejected*). The statistics of the "aligned" outputs are far from their values in the preference data.

ensemble exhibit this phenomenon: RLHF leads to rapid decreases in the length of the outputs and the fraction of outputs that contain numerical values.

These findings illustrate the tendency of different reward models to associate certain features with high reward. Policy models can exploit this association, using these features to produce outputs that are dramatically different from the reward training data, which achieve (spuriously) high reward for both individual reward models and the ensemble.

Why do reward model ensembles fail to capture uncertainty about these reward hacks? Prior work has shown that ensembles can provide good uncertainty estimates around the decision boundary, while underestimating uncertainty for examples far from the training distribution (Lakshminarayanan et al., 2017). In LM alignment, the policy can shift the output distribution away from the decision boundary to areas where all reward models erroneously extrapolate in the same manner. The same phenomenon may occur in other approaches for uncertainty estimation that are not distance-aware, such as Monte Carlo Dropout (Gal & Ghahramani, 2016) and Epistemic Neural Networks (Osband et al., 2021).

## 6 Conclusion

While reward models can improve robustness to alignment-induced distribution shift, diversity of the ensemble is crucial. Pretrain ensembles are more diverse than finetune ensembles, and therefore lead to stronger generalization. However, even pretrain ensembles are not diverse enough: for many reward hacks, all members of the ensemble agree, making the ensemble as vulnerable as its constituents. To summarize, reward model ensembles mitigate, but do not eliminate, reward hacking. Future work should examine uncertainty quantification techniques that are more robust to the type of distribution shift that occurs during alignment, particularly techniques that explicitly represent distributional shift from the preference annotations (Chu & Ghahramani, 2005; Tibshirani et al., 2019).

## 7 Reproducibility

To support reproducibility and enable further work on pretrain ensembles, we have released all $5 \times 3 = 15$ pretraining checkpoints, covering the T5-BASE, T5-LARGE, and T5-XL scales. The release can be found at https://github.com/google-deepmind/reward-ensembles. To our knowledge, this is the first release of multiple T5 checkpoints, following prior work on BERT (Sellam et al., 2021). All datasets used in this research are public. Appendix C provides all relevant hyperparameters used during RLHF.

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

# A    Numerical results for best-of-n reranking

Average agreement between reward models are shown in Tables 2-5. Autoevaluation results are shown in Table 6 and 7.

| | k | diff pretrain | same pretrain | self |
|---|---|---|---|---|
| | 1 | 0.599 | 0.599 | 0.599 |
| | 2 | 0.915 | 0.963 | 0.981 |
| | 4 | 1.155 | 1.243 | 1.275 |
| base | 8 | 1.340 | 1.462 | 1.507 |
| | 16 | 1.486 | 1.640 | 1.696 |
| | 32 | 1.605 | 1.787 | 1.854 |
| | 64 | 1.708 | 1.914 | 1.991 |
| | 1 | 0.785 | 0.785 | 0.785 |
| | 2 | 1.228 | 1.328 | 1.368 |
| | 4 | 1.556 | 1.732 | 1.805 |
| large | 8 | 1.830 | 2.069 | 2.168 |
| | 16 | 2.031 | 2.330 | 2.454 |
| | 32 | 2.203 | 2.552 | 2.697 |
| | 64 | 2.348 | 2.744 | 2.907 |
| | 1 | 0.673 | 0.673 | 0.673 |
| | 2 | 1.159 | 1.245 | 1.309 |
| | 4 | 1.513 | 1.663 | 1.780 |
| xl | 8 | 1.806 | 2.001 | 2.157 |
| | 16 | 2.023 | 2.256 | 2.449 |
| | 32 | 2.203 | 2.463 | 2.686 |
| | 64 | 2.349 | 2.631 | 2.881 |

Table 2: TL;DR best-of-n agreement.

| | k | diff pretrain | same pretrain | self |
|---|---|---|---|---|
| | 1 | 0.662 | 0.662 | 0.662 |
| | 2 | 1.081 | 1.144 | 1.178 |
| base | 4 | 1.446 | 1.560 | 1.621 |
| | 8 | 1.609 | 1.770 | 1.855 |
| | 16 | 1.776 | 1.972 | 2.078 |
| | 32 | 1.910 | 2.139 | 2.267 |
| | 1 | 0.727 | 0.727 | 0.727 |
| | 2 | 1.139 | 1.190 | 1.231 |
| large | 4 | 1.492 | 1.580 | 1.656 |
| | 8 | 1.670 | 1.791 | 1.896 |
| | 16 | 1.832 | 1.979 | 2.112 |
| | 32 | 1.962 | 2.134 | 2.291 |
| | 1 | 0.588 | 0.588 | 0.588 |
| | 2 | 1.037 | 1.079 | 1.134 |
| xl | 4 | 1.441 | 1.513 | 1.609 |
| | 8 | 1.635 | 1.731 | 1.866 |
| | 16 | 1.817 | 1.932 | 2.098 |
| | 32 | 1.963 | 2.097 | 2.293 |

Table 3: HELPFULNESS best-of-n agreement.

| | k | diff pretrain | same pretrain |
|---|---|---|---|
| base | 1 | 1.000 | 1.000 |
| | 2 | 0.811 | 0.904 |
| | 4 | 0.667 | 0.825 |
| | 8 | 0.546 | 0.756 |
| | 16 | 0.447 | 0.695 |
| | 32 | 0.366 | 0.637 |
| | 64 | 0.303 | 0.589 |
| large | 1 | 1.000 | 1.000 |
| | 2 | 0.780 | 0.886 |
| | 4 | 0.616 | 0.794 |
| | 8 | 0.497 | 0.720 |
| | 16 | 0.394 | 0.651 |
| | 32 | 0.319 | 0.593 |
| | 64 | 0.260 | 0.546 |
| xl | 1 | 1.000 | 1.000 |
| | 2 | 0.781 | 0.859 |
| | 4 | 0.618 | 0.743 |
| | 8 | 0.503 | 0.655 |
| | 16 | 0.400 | 0.567 |
| | 32 | 0.323 | 0.497 |
| | 64 | 0.262 | 0.433 |

Table 4: TL;DR top 1 agreement.

| | k | diff pretrain | same pretrain |
|---|---|---|---|
| base | 1 | 1.000 | 1.000 |
| | 2 | 0.805 | 0.885 |
| | 4 | 0.650 | 0.789 |
| | 8 | 0.506 | 0.695 |
| | 16 | 0.406 | 0.620 |
| | 32 | 0.318 | 0.548 |
| large | 1 | 1.000 | 1.000 |
| | 2 | 0.810 | 0.874 |
| | 4 | 0.656 | 0.766 |
| | 8 | 0.522 | 0.668 |
| | 16 | 0.413 | 0.579 |
| | 32 | 0.324 | 0.506 |
| xl | 1 | 1.000 | 1.000 |
| | 2 | 0.812 | 0.860 |
| | 4 | 0.666 | 0.746 |
| | 8 | 0.536 | 0.635 |
| | 16 | 0.436 | 0.547 |
| | 32 | 0.345 | 0.466 |

Table 5: HELPFULNESS top-1 agreement.

| scale | ensemble | method | reward | win rate |
|---|---|---|---|---|
| base | finetune | mean | $-0.220$ | 0.700 |
| | | mean minus std | $-0.186$ | 0.703 |
| | | median | $-0.231$ | 0.700 |
| | | min | $-0.177$ | 0.710 |
| | pretrain | mean | $-0.130$ | 0.721 |
| | | mean minus std | $-0.086$ | 0.731 |
| | | median | $-0.155$ | 0.715 |
| | | min | $-0.086$ | 0.727 |
| | single RM | single RM | $-0.244$ | 0.685 |
| large | finetune | mean | 0.342 | 0.814 |
| | | mean minus std | 0.343 | 0.816 |
| | | median | 0.309 | 0.809 |
| | | min | 0.348 | 0.813 |
| | pretrain | mean | 0.549 | 0.850 |
| | | mean minus std | 0.513 | 0.847 |
| | | median | 0.510 | 0.846 |
| | | min | 0.475 | 0.841 |
| | single RM | single RM | 0.280 | 0.792 |
| xl | finetune | mean | 0.695 | 0.873 |
| | | mean minus std | 0.644 | 0.872 |
| | | median | 0.625 | 0.867 |
| | | min | 0.638 | 0.868 |
| | pretrain | mean | 0.831 | 0.900 |
| | | mean minus std | 0.781 | 0.895 |
| | | median | 0.757 | 0.889 |
| | | min | 0.735 | 0.883 |
| | single RM | single RM | 0.585 | 0.853 |

Table 6: TL;DR BoN ($n = 64$) autoeval results (T5-XXL fine-tuned evaluator).

| scale | ensemble | method | reward | win rate |
|---|---|---|---|---|
| base | finetune | mean | 0.635 | 0.741 |
| | | mean minus std | 0.615 | 0.735 |
| | | median | 0.627 | 0.738 |
| | | min | 0.604 | 0.725 |
| | pretrain | mean | 0.691 | 0.752 |
| | | mean minus std | 0.661 | 0.748 |
| | | median | 0.683 | 0.749 |
| | | min | 0.624 | 0.741 |
| | single RM | single RM | 0.600 | 0.727 |
| large | finetune | mean | 0.778 | 0.776 |
| | | mean minus std | 0.758 | 0.772 |
| | | median | 0.771 | 0.770 |
| | | min | 0.738 | 0.766 |
| | pretrain | mean | 0.847 | 0.792 |
| | | mean minus std | 0.802 | 0.779 |
| | | median | 0.813 | 0.784 |
| | | min | 0.770 | 0.776 |
| | single RM | single RM | 0.730 | 0.759 |
| xl | finetune | mean | 0.884 | 0.805 |
| | | mean minus std | 0.837 | 0.788 |
| | | median | 0.859 | 0.790 |
| | | min | 0.814 | 0.788 |
| | pretrain | mean | 0.932 | 0.816 |
| | | mean minus std | 0.876 | 0.797 |
| | | median | 0.892 | 0.798 |
| | | min | 0.858 | 0.792 |
| | single RM | single RM | 0.811 | 0.779 |

Table 7: HELPFULNESS BoN ($n = 32$) autoeval results (T5-XXL fine-tuned evaluator).

# B  Numerical results for RLHF

Full RLHF numerical results for HELPFULNESS and TL;DR are shown in Table 8 and Table 9.

| Ensemble | Method | $\lambda$ | Reward xl | Reward large | Reward base |
|---|---|---|---|---|---|
| ft | mean | 0.010 | 2.562 | | |
| ft | mean | 0.025 | 2.476 | 2.204 | 0.041 |
| ft | mean | 0.050 | 2.148 | 2.089 | 1.503 |
| ft | mean | 0.100 | 1.652 | 1.591 | 1.497 |
| ft | mean | 0.150 | 1.328 | 1.258 | 1.212 |
| ft | mean | 0.200 | 1.079 | 1.032 | 0.980 |
| ft | mean | 0.300 | 0.764 | 0.688 | 0.666 |
| ft | mean subtract std | 0.010 | 2.478 | | |
| ft | mean subtract std | 0.025 | 2.401 | 2.188 | 0.240 |
| ft | mean subtract std | 0.050 | 2.118 | 1.978 | 1.585 |
| ft | mean subtract std | 0.100 | 1.620 | 1.525 | 1.432 |
| ft | mean subtract std | 0.150 | 1.315 | 1.207 | 1.152 |
| ft | mean subtract std | 0.200 | 1.089 | 0.998 | 0.949 |
| ft | mean subtract std | 0.300 | 0.746 | 0.667 | 0.648 |
| ft | median | 0.010 | 2.466 | | |
| ft | median | 0.025 | 2.425 | 2.088 | 0.153 |
| ft | median | 0.050 | 2.154 | 2.051 | 1.445 |
| ft | median | 0.100 | 1.662 | 1.585 | 1.489 |
| ft | median | 0.150 | 1.318 | 1.255 | 1.197 |
| ft | median | 0.200 | 1.096 | 1.022 | 0.976 |
| ft | median | 0.300 | 0.750 | 0.699 | 0.676 |
| pt | mean | 0.010 | 2.651 | | |
| pt | mean | 0.025 | 2.551 | 2.293 | 1.220 |
| pt | mean | 0.050 | 2.196 | 2.099 | 1.750 |
| pt | mean | 0.100 | 1.724 | 1.498 | 1.506 |
| pt | mean | 0.150 | 1.319 | 1.225 | 1.191 |
| pt | mean | 0.200 | 1.106 | 1.014 | 0.958 |
| pt | mean | 0.300 | 0.759 | 0.643 | 0.680 |
| pt | mean subtract std | 0.010 | 2.688 | | |
| pt | mean subtract std | 0.025 | 2.529 | 2.293 | 0.636 |
| pt | mean subtract std | 0.050 | 2.167 | 2.025 | 1.696 |
| pt | mean subtract std | 0.100 | 1.695 | 1.450 | 1.342 |
| pt | mean subtract std | 0.150 | 1.301 | 1.188 | 1.117 |
| pt | mean subtract std | 0.200 | 1.099 | 1.007 | 0.932 |
| pt | mean subtract std | 0.300 | 0.732 | 0.688 | 0.641 |
| pt | median | 0.010 | 2.611 | | |
| pt | median | 0.025 | 2.540 | 2.383 | 1.166 |
| pt | median | 0.050 | 2.141 | 2.064 | 1.662 |
| pt | median | 0.100 | 1.674 | 1.488 | 1.537 |
| pt | median | 0.150 | 1.365 | 1.181 | 1.220 |
| pt | median | 0.200 | 1.117 | 0.973 | 1.014 |
| pt | median | 0.300 | 0.743 | 0.669 | 0.661 |
| single RM | n/a | 0.010 | 2.245 | | |
| single RM | n/a | 0.025 | 2.321 | 1.511 | -0.349 |
| single RM | n/a | 0.050 | 2.024 | 1.834 | 1.028 |
| single RM | n/a | 0.100 | 1.594 | 1.478 | 1.432 |
| single RM | n/a | 0.150 | 1.297 | 1.194 | 1.148 |
| single RM | n/a | 0.200 | 1.069 | 0.988 | 0.937 |
| single RM | n/a | 0.300 | 0.759 | 0.661 | 0.636 |

Table 8: HELPFULNESS RLHF numerical results.

| Ensemble | Method | $\lambda$ | Reward xl | Reward large | Reward base |
|---|---|---|---|---|---|
| ft | mean | 0.010 | 2.356 | 1.562 | -1.310 |
| ft | mean | 0.030 | 2.088 | 1.659 | -0.456 |
| ft | mean | 0.100 | 1.073 | 0.779 | -0.389 |
| ft | mean | 0.300 | -0.217 | -0.380 | -0.785 |
| ft | mean | 0.500 | -0.707 | -0.785 | -0.964 |
| ft | mean subtract std | 0.010 | 2.171 | 1.579 | -1.185 |
| ft | mean subtract std | 0.030 | 1.760 | 1.533 | -0.392 |
| ft | mean subtract std | 0.100 | 0.811 | 0.658 | -0.359 |
| ft | mean subtract std | 0.300 | -0.303 | -0.409 | -0.777 |
| ft | mean subtract std | 0.500 | -0.735 | -0.791 | -0.960 |
| ft | median | 0.010 | 2.206 | 1.480 | -1.939 |
| ft | median | 0.030 | 1.939 | 1.596 | -0.509 |
| ft | median | 0.100 | 0.955 | 0.809 | -0.376 |
| ft | median | 0.300 | -0.265 | -0.370 | -0.789 |
| ft | median | 0.500 | -0.728 | -0.788 | -0.963 |
| pt | mean | 0.010 | 2.366 | 2.037 | -0.817 |
| pt | mean | 0.030 | 1.997 | 1.852 | -0.343 |
| pt | mean | 0.100 | 0.964 | 0.858 | -0.366 |
| pt | mean | 0.300 | -0.309 | -0.377 | -0.776 |
| pt | mean | 0.500 | -0.744 | -0.786 | -0.962 |
| pt | mean subtract std | 0.010 | 2.398 | 2.019 | -0.957 |
| pt | mean subtract std | 0.030 | 1.997 | 1.710 | -0.250 |
| pt | mean subtract std | 0.100 | 1.002 | 0.768 | -0.328 |
| pt | mean subtract std | 0.300 | -0.277 | -0.357 | -0.767 |
| pt | mean subtract std | 0.500 | -0.752 | -0.774 | -0.958 |
| pt | median | 0.010 | 2.431 | 2.009 | -0.868 |
| pt | median | 0.030 | 2.030 | 1.903 | -0.317 |
| pt | median | 0.100 | 1.086 | 0.850 | -0.347 |
| pt | median | 0.300 | -0.308 | -0.388 | -0.778 |
| pt | median | 0.500 | -0.746 | -0.792 | -0.962 |
| single RM | n/a | 0.010 | 1.728 | 1.429 | -1.784 |
| single RM | n/a | 0.030 | 1.590 | 1.511 | -0.458 |
| single RM | n/a | 0.100 | 0.787 | 0.758 | -0.397 |
| single RM | n/a | 0.300 | -0.299 | -0.387 | -0.783 |
| single RM | n/a | 0.500 | -0.736 | -0.791 | -0.966 |

Table 9: TL;DR RLHF numerical results.

## C  Hyperparameters

We provide the hyperparameters for reward model and RLHF training in Table 10 and Table 11. For reward models, we use the validation set to choose the best checkpoint along training. For RLHF, we take the last checkpoint.

## D  Additional results

| Task | Parameter | value |
|------|-----------|------:|
| Helpfulness | Learning rate | $10^{-4}$ |
|  | Learning schedule | Constant (linear warm-up) |
|  | Warm-up steps | 500 |
|  | Dropout | 0.05 |
|  | Batch size | 64 |
|  | $\eta$ (regularization coefficient) | 0.01 |
| TL;DR | Learning rate | $10^{-4}$ |
|  | Learning schedule | Constant (linear warm-up) |
|  | Warm-up steps | 1000 |
|  | Dropout | 0.05 |
|  | Batch size | 32 |
|  | $\eta$ (regularization coefficient) | 0.01 |
| XSum/NLI | Learning rate | base/large: $10^{-3}$, xl: $3 \cdot 10^{-3}$ |
|  | Learning schedule | constant |
|  | Warm-up steps | - |
|  | Dropout | 0.01 |
|  | Batch size | base/large: 128, xl: 32 |

Table 10: Hyper-parameters for reward model training.

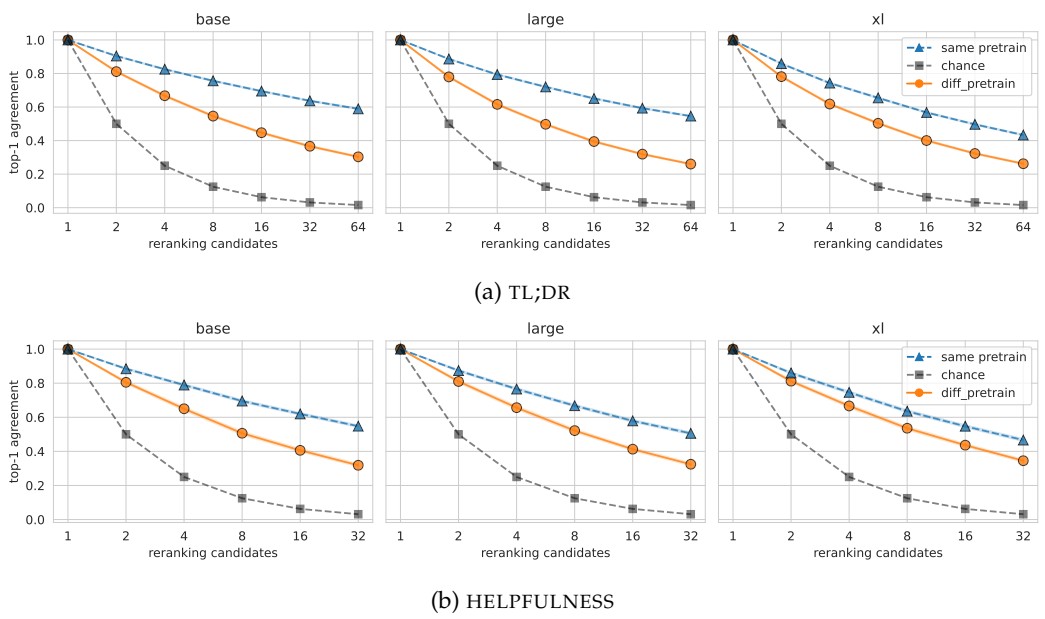

(a) TL;DR

(b) HELPFULNESS

Figure 8: Agreement of the top-ranked output between reward models that do (crosses) and do not (circles) share pretraining seeds. Underspecification of reward models directly affects the behavior of the aligned policy. Chance agreement is $1/n$.

| Task | Parameter | value |
|------|-----------|------:|
| Helpfulness | Policy learning rate | $5 \cdot 10^{-6}$ |
| | Value learning rate | $10^{-5}$ |
| | Learning schedule | Constant (linear warm-up) |
| | Training steps | 20000 |
| | Warm-up steps | 2000 |
| | Batch size | base/large: 32, xl: 16 |
| | input length | 1024 |
| | output length | 256 |
| | $\lambda$ | [0.01, 0.025, 0.05, 0.1, 0.15, 0.2, 0.3] |
| TL;DR | Policy learning rate | $5 \cdot 10^{-6}$ |
| | Value learning rate | $10^{-5}$ |
| | Learning schedule | Constant (linear warm-up) |
| | Training steps | 20000 |
| | Warm-up steps | 2000 |
| | Batch size | 32 |
| | input length | 1024 |
| | output length | 128 |
| | $\lambda$ | [0.01, 0.03, 0.1, 0.3, 0.5] |
| XSum/NLI | Policy learning rate | $5 \cdot 10^{-6}$ |
| | Value learning rate | $10^{-5}$ |
| | Learning schedule | Constant (linear warm-up) |
| | Training steps | 20000 |
| | Warm-up steps | 2000 |
| | Batch size | 32 |
| | input length | 1024 |
| | output length | 64 |
| | $\lambda$ | [0.01, 0.03, 0.05, 0.1, 0.3, 0.5] |

Table 11: Hyper-parameters for RLHF.

| Task | Prompt | Output |
|------|--------|--------|
| TL;DR | I'll try to keep this short! **Background** * I've always been an on again/off again (very casual!) jogger, typically doing 3 - 5 k * My knees have always been finicky, and I went to a physio who thought I had "runner's knee" [...] I had my baby a year ago, so all in all I haven't run for about 1.5 years [...] However, when I run even the tinyiest bit, or even go on a long walk or a hike, my pelvis gets very sore and tight [...] | I think pregnancy messed with my body, now I can't even run even the smallest amount without pain in my pelvis and knees. I'm fairly certain the problem isn't just that I'm completely out of shape. |
| Helpfulness | **Human**: Assistant, can you help me find books that have really popular antagonists? **Assistant**: Sure! I know many books that feature good antagonists, such as those in the Lord of the Rings books. **Human**: Who specifically from Lord of the Rings? **Assistant**: | Well, Sauron, for example. If you want, I can explain who he was. |
| XSum/NLI | The ex-Reading defender denied fraudulent trading charges relating to the Sodje Sports Foundation - a charity to raise money for Nigerian sport. Mr Sodje, 37, is jointly charged with elder brothers Efe, 44, Bright, 50 and Stephen, 42. Appearing at the Old Bailey earlier, all four denied the offence. The charge relates to offences which allegedly took place between 2008 and 2014. Sam, from Kent, Efe and Bright, of Greater Manchester, and Stephen, from Bexley, are due to stand trial in July. They were all released on bail. | Former Premier League footballer Sam Sodje has appeared in court alongside three brothers accused of charity fraud. |

Table 12: Prompt-output pairs for the three benchmarks we consider. See §2.3.

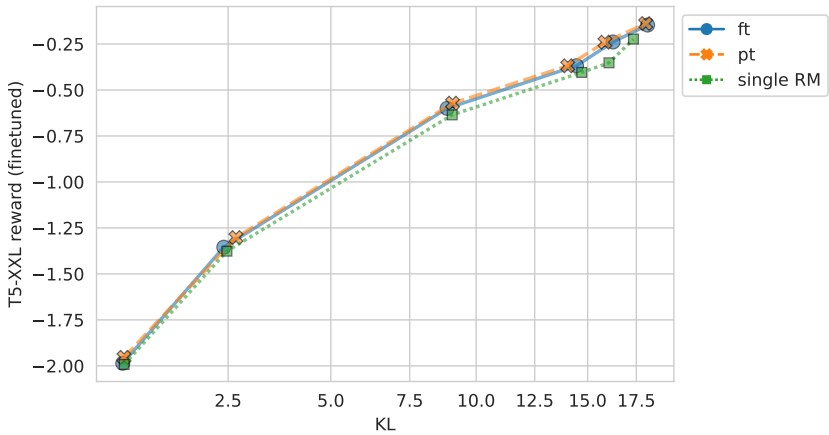

Figure 9: XSUM/NLI KL-reward tradeoff for pretrain ensembles, finetune ensembles, and individual models. Reward is measured with T5-XXL. Both pretrain and finetune ensembles slightly improve over individual models.

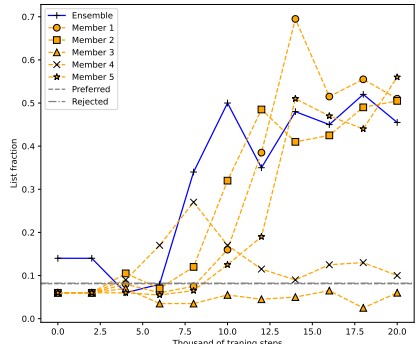

(a) HELPFULNESS. Fraction of answers containing lists (as matched by a regular expression).

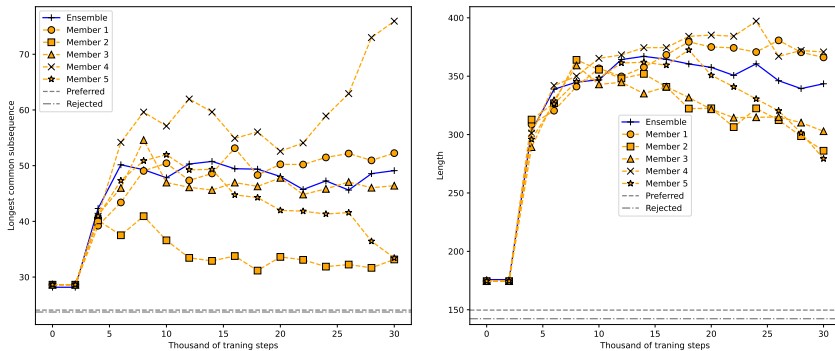

(b) TL;DR. Left: extractiveness, as measured by average longest common substring between the summary and the context document. Right: length.

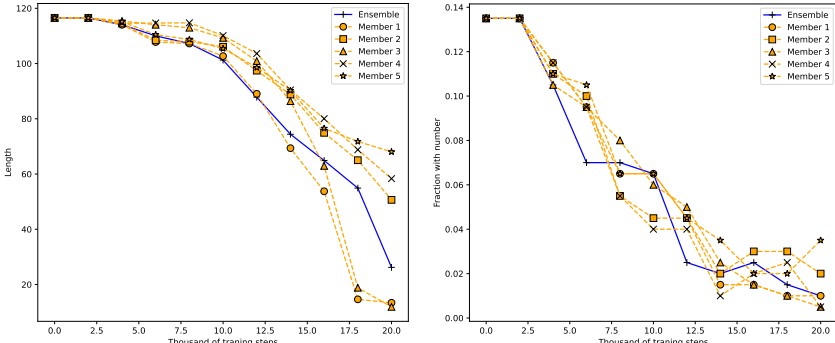

(c) XSUM/NLI. Left: length. Right: specificity, as measured by fraction of numerical tokens in the output.

Figure 10: Limitations of reward model ensembles. The x-axis is number of RLHF steps, the y-axis plots different statistics of the average validation output at that step, and the curves correspond to the pretrain ensemble (solid blue) and its members (dashed orange). For preference data, we plot the same statistics conditioned on the preference data label (*Preferred* vs. *Rejected*). On HELPFULNESS ($\lambda = 0.05$, top), the ensemble tends to return a list of items. On TL;DR (center, $\lambda = 0.01$), summaries become longer and copy longer spans from the original document. For XSUM/NLI ($\lambda = 0.03$, bottom), responses are short and less specific, as measured by lack of numerical information. In HELPFULNESS and TL;DR, the statistics of the "aligned" outputs are far from their values in the preference data.

