# OpenReview forum: "Helping or Herding? Reward Model Ensembles Mitigate but do not Eliminate Reward Hacking"
_colmweb.org/COLM/2024/Conference — COLM_

### Official Review · Reviewer_Aua6 · 2024-05-03

**Rating:** 7
**Confidence:** 4
**Ethics Flag:** 1

**Summary:**

The commonly used preference alignment approach involves training a reward model and optimizing the output of the LLM against this reward model. The effectiveness of the reward model directly impacts the preference alignment. Recently, some studies have attempted to improve reward models through the use of a reward model ensemble technique, which can help avoid the phenomenon of reward hacking. This paper further explores the use of the reward model ensemble in preference alignment and presents some insightful conclusions based on experiments in the summarization and dialogue tasks.

**Questions To Authors:**

1. This seems to be a typing error in the sentence: “PALM-2 on the two possible orderings....”
2. Have you tried using list-wise data to train the reward models? If so, is this phenomenon the same as with pair-wise data?
3. Has the reward score of each reward model been normalized or undergone other preprocessing steps?

**Reasons To Accept:**

1. This paper is easy to read, and the conclusions are clear. This paper explores RM ensemble through reinforcement learning and re-ranking, and evaluates the effects of the data distribution and reward model type.
2. Building high-quality reward signals for preference alignment training is crucial. This paper can help in understanding the reward behavior and better constructing the reward ensemble, thus outputting higher-quality reward signals for the preference alignment.

**Reasons To Reject:**

1. As shown in Table 1, although T5-XXL can outperform the best T5-XL model on each task, using the win-rate computed by T5-XXL as an evaluation metric for each task is inadequate. One concern is that T5-XXL exhibits a significant amount of reward hacking behavior, similar to that of the previously used reward models, such as T5-Large and T5-XL. Therefore, it is recommended to also use other metrics (e.g., BARTScore and GPT-4 Win Rate) to further evaluate the policy model.

2. Given OpenAI's reward scaling law, one might wonder if a larger reward model could effectively mitigate reward hacking. It raises a natural question: Is it more beneficial to use a larger reward model, or consider an ensemble of smaller reward models? Furthermore, what would be the impact of introducing more larger reward model into the reward model collection for ensembling? However, these questions are not discussed in this paper.

3. The reward model ensemble discussed in this paper overlooks a significant issue: the differences in score margins between the reward models intended for ensembling. These differences can lead to error results for the reward model ensemble. For instance, consider two samples, a and b. The first reward model assigns scores of -5 and -6, respectively, while the second reward model assigns scores of 3 and 20, respectively. When applying the reward ensemble for re-ranking, the negative score assigned by the first reward model to sample b can be easily overlooked. However, we cannot guarantee that the margin measure is consistent across all reward models. There is a possibility that the first reward model deems sample b to be significantly inferior to sample a, despite only a one-point difference in their scores.

---

> ### Author Rebuttal · Authors · 2024-05-30
>
> Thanks for the careful consideration of our submission!
>
> **T5-XXL rewards the same hacks as the smaller scale models**: we agree, which is why we also evaluate the win rate with a prompted auto-eval model in Figure 6. This autoeval model is not trained on any preference data. The issue is discussed on the bottom of page 7.
>
> **Comparison of a larger reward model to an ensemble of smaller RMs**: To a limited extent, a comparison can be extracted from our results because the T5-XL models are roughly four times larger than the T5-large models (3B vs 770M parameters), making the T5-large ensemble comparable in parameter count to T5-XL. According to the fine-tuned autoeval model, performance of the two setups is similar. For example in BoN, a T5-large pretrain ensemble gets win rates of .85 (tldr) and .79 (helpfulness), vs .85 and .78 for a single T5-XL model. However, our results demonstrate that ensembling can *further* improve the performance of a policy trained with a T5-XL reward model. Ensembles across multiple RM scales is an intriguing idea for future work.
>
> **Potential "difference in score margins between the [RMs] used for ensembling."** There are two potential issues: scale and offset. As discussed in section 2.1, the Bradley-Terry model, which defines the RM training objective, is undetermined with regard to offset (r=[-5, -7] is equivalent to r=[107, 105]). This is why we add an additional regularization term to "center" the rewards around zero (equation 2). However, the BT model is *not* undetermined with regard to scale: an assignment of r=[-1, 1] is not equivalent to r=[-10, 10] from the perspective of the reward model training objective. Therefore we do not expect RMs trained on the same data to produce rewards of significantly different magnitudes.
>
> Questions:
> - Thanks for catching the typo in the final paragraph of section 2, we will correct it.
> - The idea of training the RMs on listwise data is an interesting proposed extension of the RLHF methodology, but we do not anticipate significant differences in terms of reward hacking. Listwise preferences are only available in limited settings.
> - Regarding reward score normalization, please see the discussion above about section 2.1 of the paper. The scores are not directly postprocessed, but the objective is changed to remove the shift-invariance of the BT model.

---

> > ### Comment · Reviewer_Aua6 · 2024-06-04
> >
> > Thank you for your response. Some of my concerns have been addressed. I'd like to see this submission accepted.

---

### Official Review · Reviewer_jRNy · 2024-05-09

**Rating:** 7
**Confidence:** 4
**Ethics Flag:** 1

**Summary:**

This paper investigates the problem of reward hacking in language model alignment using reward models. The authors show that reward models trained on the same data but with different random seeds can disagree significantly on out-of-distribution data generated by the policy during the alignment process. This underspecification of reward models propagates to the aligned policy, making it highly tuned to the specific reward model used during training. To mitigate this issue, the authors propose using reward model ensembles, particularly "pretrain ensembles" where each member is pretrained with a different random seed. Experiments on several language tasks demonstrate that pretrain ensembles are more robust than individual reward models or "finetune ensembles" where members only differ in the finetuning seed. However, the authors find that even pretrain ensembles do not eliminate reward hacking, as they can fail to capture uncertainty and penalize certain undesirable behaviors that are incentivized by all ensemble members.

**Reasons To Accept:**

- The paper analyzes the problem of underspecification in reward modeling, providing empirical evidence of its prevalence and downstream effects on the aligned policy.
- The proposed solution of pretrain ensembles is well-motivated and evaluated on multiple tasks with different alignment strategies (best-of-n and RLHF). The results convincingly demonstrate the advantages of pretrain diversity.
- The authors identify important failure modes of reward ensembles through qualitative analysis, showing that they don't fully solve reward hacking. This provides a balanced and insightful perspective on the limitations of the proposed approach.

**Reasons To Reject:**

- Pretraining multiple reward models is computationally expensive, especially at larger model scales. The paper doesn't discuss the trade-off between ensemble diversity and training cost.
- I think RMs might not be significantly different from each other, even though they were trained with different seeds. To promote much more diversity among the RMs, a better strategy for the RMs’ training would be explored, e.g., training each RM with a fraction of preference datasets like K-fold cross-validation or ensembles of RMs trained with different alignment datasets.
- They did not compare the ensemble of small RMs and a single larger RM in terms of reward overoptimization. In some cases, training a single large RM might be more affordable.

---

> ### Author Rebuttal · Authors · 2024-05-30
>
> Thanks for the careful consideration of our submission! We were happy that you found the results convincing and the analysis insightful.
>
> Computational expense of multiple pretrains: we agree, with two caveats. First, the effort to create a pretrain ensemble could be amortized across many reward models (as we have done in our own experiments), and might also be useful for instruction tuning. Second, our work motivates the search for more sophisticated strategies that may yield similar benefits to full-scale ensembling at lower cost, such as, e.g. LoRA ensembles [1, 2] and weight-space averaging [3] after projection into a shared weight space [4].
>
> Ensemble diversity: as shown in Figure 3 the ensemble makes diverse predictions even before RLHF. Fine-tune ensembles have an average rank correlation between .45 and .62 on five outputs from the same reference model, and the average rank correlation for pretrain ensembles is significantly lower. That said, we agree that more could be done to encourage diversity in finetune ensembles, and the reviewer's suggestion of a bootstrap ensemble seems particularly promising. Our focus on random seed variability aligns with our emphasis on underspecification on deep networks from in-distribution training data [5], but we agree that other sources of variability are of interest.
>
> Comparison of an ensemble of small RMs vs a single larger RM: to a limited extent, this comparison can be extracted from our results because the T5-XL models are roughly four times larger than the t5-large models (3B vs 770M parameters), making the T5-large ensemble comparable in parameter count to T5-XL. According to the fine-tuned autoeval model, performance of the two setups is similar. For example in BoN, a T5-large pretrain ensemble gets win rates of .85 (tldr) and .79 (helpfulness), vs .85 and .78 for a single T5-XL model. Morevoer, our results demonstrate that ensembling can *further* improve the performance of a policy trained with a T5-XL reward model. As for affordability, a reward ensemble of size 5xN may enable faster inference during RLHF than a single reward model of size 5N, since inference over the reward models can be easily parallelized; similarly, parallelization may make it easier to pretrain five LMs of size N than a single LM of size 5N.
>
> [1] https://arxiv.org/abs/2310.00035
> [2] https://arxiv.org/abs/2405.14438v1
> [3] https://arxiv.org/abs/2306.04488
> [4] https://arxiv.org/abs/2209.04836
> [5] https://arxiv.org/abs/2011.03395

---

> > ### Comment · Reviewer_jRNy · 2024-06-04
> >
> > Thank you for the response. I'll keep my positive score.

---

### Official Review · Reviewer_2M6i · 2024-05-11

**Rating:** 7
**Confidence:** 4
**Ethics Flag:** 1

**Summary:**

This paper studies the use of reward models in aligning language model with human preference with a focus on addressing the reward hacking using reward model ensemble. In RLHF, the policy may exploit the model error to achieve a good performance measured by the imperfect reward model, which is usually called reward hacking or reward over optimization. The authors investigate how well reward model ensemble can mitigate this issue during training and inference. Further, the author also study why does reward model ensemble will fail sometimes. The key findings in this paper are:

1. Reward models are underspecified, which leads to different rewards or rankings when tested on out-of-distribution data.
2. Overoptimization occurs when alignment to one reward model does not improve reward as measured by another.
3. Reward ensembles, especially pretrained ensembles (with different random seeds), help mitigate overoptimization and improve generalization.
4. Reward hacking is not completely eliminated even with reward model ensemble. This is because the reward model ensemble will underestimate the uncertainty if it's far away from the data distribution.

**Reasons To Accept:**

This paper investigates a critical issue: aligning language models with human preferences. It provides a thorough examination of how reward model ensembles can mitigate over-optimization. The finding that ensembles composed of members pre-trained using different seeds generalize better offers valuable insights for practitioners, albeit unsurprising.

Overall, the paper is well-organized, and the experimental results solidly support each argument, making it an enjoyable read.

**Reasons To Reject:**

The new insights from the paper may seem limited after reading it. The use of an ensemble of uncertainty quantification methods to mitigate reward hacking is not novel in the literature, as many reinforcement learning papers, especially those on offline RL, adopt a similar approach to address the reward hacking problem.

The authors hypothesize that the same phenomenon will occur in other approaches for uncertainty estimation that are not aware of distance. I wonder if the authors have tried using SNGP on top of the reward model to make it distance-aware when quantifying uncertainty. If the hypothesis holds true, it should perform better than an ensemble in mitigating reward hacking.

Additionally, to validate the above conjecture, it would be interesting to see how increasing the number of ensembles affects performance. If the issue is largely due to being "distance unaware," increasing the number of ensembles should not be helpful beyond a certain point.

Implementing this may require significant resources, and it might be impractical to obtain several large pretrained models with different seeds.

---

> ### Author Rebuttal · Authors · 2024-05-30
>
> Thanks for the feedback on our submission. We're glad that you found reading the paper to be enjoyable!
>
> Regarding insights: Our work is indeed inspired by past work on using ensembles in offline RL, typically in the dynamics model. However, applying ensembles for LLM alignment is more nuanced due to the interaction between the ensemble members and the process of pretraining. A key result of our work is that pretrain ensembles are more effective than finetune ensembles, a new finding that is important for better understanding the persistently difficult problem of combatting reward hacking during LLM alignment.
>
> Distance-aware uncertainty quantification: We agree that distance-aware uncertainty quantification could be a promising next step for uncertainty-aware reward modeling and appreciate the suggestion of SNGP. To our knowledge, however, SNGP has only been applied at the scale of T5-base [1] or with a small number of training examples [6, concurrent with our work]. We would certainly be interested in investigating how to leverage SNGP with modern LLM and practical training sets.
>
> Size of ensemble: In pilot studies in the best-of-n framework, we did not find a large effect of the ensemble size, although we agree that more comprehensive evaluation is needed. As the reviewer notes, experiments on ensemble size are computationally intensive, particularly for pretrain ensembles. However, we can explore the effect of ensemble size on fine-tune ensembles in the best-of-n setting, and will evaluate 10-member ensembles in the best-of-n setting for the next version of the paper.
>
> Computational expense of multiple pretrains: we agree, with two caveats. First, the effort to create a pretrain ensemble could be amortized across many reward models (as we have done in our own experiments on summarization and dialogue), and might also be useful for instruction tuning. Second, our work motivates the search for more sophisticated strategies that may yield similar benefits to full-scale ensembling at lower cost, such as, e.g. LoRA ensembles [2, 3] and weight-space averaging [4] after projection into a shared weight space [5].
>
> [1] https://arxiv.org/abs/2207.07411
> [2] https://arxiv.org/abs/2310.00035
> [3] https://arxiv.org/abs/2405.14438v1
> [4] https://arxiv.org/abs/2306.04488
> [5] https://arxiv.org/abs/2209.04836
> [6] https://arxiv.org/pdf/2403.05171v1

---

> > ### Comment · Reviewer_2M6i · 2024-06-05
> >
> > Thanks for the authors' response. I don't have additional questions / concerns.

---

### Official Review · Reviewer_kjNj · 2024-05-11

**Rating:** 6
**Confidence:** 4
**Ethics Flag:** 1

**Summary:**

This paper looks at the issue of reward hacking with reward model ensembles. The datasets, models, and training methods are all from previous, except for how different individual reward models are trained and ensembled together. Experiments first show that different reward models predict similar scores for in-distribution data, but disagree substantially for out-of-distribution data. Experiments with reward ensembles show that ensembles usually provide higher performance in best-of-n reranking as well as RLHF finetuning. However, reward hacking is still observed even when reward model ensembles are used in RLHF.

Strengths:
- The issue of reward hacking and the prevention of it are important problems.
- The experiments presented in the paper are thorough and support most of the claims in the paper.
- Analysis and discussion provide good insights into the issue.

Weaknesses:
- Lack of human evaluation. The only human evaluation is qualitative in nature.
- Ensembling is a well known method to reduce overreliance of spurrious features. The paper does not contribute to the method itself.

**Reasons To Accept:**

See strengths.

**Reasons To Reject:**

See weaknesses.

---

> ### Author Rebuttal · Authors · 2024-05-30
>
> Thanks for the feedback on our submission! We were pleased to read that you found that the experiments and analysis were thorough and insightful.
>
> While we do not offer a new ensembling method, our paper sheds new light on the relevance of ensembling in the pretraining era, and in particular for LLM alignment. First, it is necessary to revisit the ideas behind ensembling given the widespread use of pretraining as an initial point for the ensemble members. To our knowledge, we are the first to compare ensembles within and across pretrains. Second, we are among the first to consider the relevance of ensembling to the phenomenon of reward hacking in RLHF, which, as we argue in the paper, is related to but not coextensive with overfitting.
>
> Regarding the lack of human evaluation, we agree that this would strengthen the empirical results of the paper and hope to explore it in future work. In particular, it would be good to ensure that the gains offered by reward ensembles in automated evaluations also hold up in human evaluations. This evaluation challenge is shared by a large number of recent studies (e.g., all those using [1] for evaluation) where language models generate long responses, which render human evaluation prohibitively expensive. Due to the cost and time-intensity, unfortunately we cannot commit to performing such an evaluation for the next version of this paper. We emphasize that prompt-based and fine-tuned autoeval models tell a similar story about the effects of pretrain and finetune ensembles, increasing our confidence that these effects are real.
>
> [1] AlpacaFarm: A Simulation Framework for Methods that Learn from Human Feedback. Dubious et al., NeurIPS 2023.

---

### Decision · Program_Chairs · 2024-07-10

**Decision:**

Accept

**Comment:**

## Summary of the paper

The paper studies the effect of ensembling reward models (RMs) in the context of RLHF for LLMs. Specifically, the authors show:
- RMs are underspecified: when trained with different random seeds, they give different rankings on OOD inputs.
- Ensembling over pretrains leads to better uncertainty estimates than ensembling over finetunes.
- Ensembling RMs can lead to better results in RLHF and in best-of-N reranking.
- Ensembling does not remove all reward hacks, as some are consistent across retrains.

## Strengths

- This is a high-quality study, with very carefully designed experiments. The authors consider a collection of models of different sizes, complementary evaluations (Palm 2 win-rate + T5-XXL reward), multiple tasks.
- The paper is well-written, with good structure, writing and figures.
- The paper studies a question which is very relevant in practice. Improvements to RMs can lead to major qualitative improvements in current LLMs.
- Ensembling is also a pretty generic and fundamental tool for improving models, so I expect the results are likely to generalize to other setup variations.
- Results are overall positive and promising.
- Section 5 on failure modes is very interesting.

## Weaknesses

- Ensembling is a generic method, and similar studies with similar conclusions have been conducted many times in other settings (RL, computer vision, ...)
- Given the prior studies in other domains, the qualitative results are not very surprising
- Only very limited results on training an ensemble vs training a larger RM. Ultimately, it would be interesting to see a collection of scaling laws for different ways of growing the RM: growing the ensemble and growing individual models.
- No human evaluations of the quality of the policy model trained with the ensemble RM.

## Question

I am not sure if I missed it, but I would be quite curious to know if reusing the same pretrain for the policy and RM in RLHF is worse than using different pretrains?

## Recommendation

I think this is a good paper which would be a valuable contribution to the conference.